# Adherence to the modified Barcelona Clinic Liver Cancer guidelines: Results from a high-volume liver surgery center in East Asias

Yi-Hao Yen[1]*, Yu-Fan Cheng[2], Jing-Houng Wang[1], Chih-Che Lin[3], Chien-Hung Chen[1], Chih-Chi Wang[3]

**1** Division of Hepatogastroenterology, Department of Internal Medicine, Kaohsiung Chang Gung Memorial Hospital and Chang Gung University College of Medicine, Kaohsiung, Taiwan, **2** Department of Diagnostic Radiology, Liver Transplantation Center, Kaohsiung Chang Gung Memorial Hospital, Chang Gung University College of Medicine, Kaohsiung, Taiwan, **3** Department of Surgery, Liver Transplantation Center, Kaohsiung Chang Gung Memorial Hospital, Kaohsiung, Taiwan

\* casselyen@yahoo.com.tw

## Abstract

### Background and aims

The Barcelona Clinic Liver Cancer (BCLC) staging system is the most widely applied staging system for hepatocellular carcinoma (HCC) and is recommended for treatment allocation and prognostic prediction. The BCLC guidelines were modified in 2018 to indicate that Child-Pugh A without any ascites is essential for all stages except stage D. This study sought to provide a description of patients with HCC treated at a high-volume liver surgery center in Taiwan where referral is not needed and all treatment modalities are available and reimbursed by the National Health Insurance program. As such, certain variables that could modulate treatment decisions in clinical practice, including financial constraints, the availability of treatment procedures, and the expertise of the hospital, could be excluded. The study further sought to evaluate the adherence to the modified BCLC guidelines.

### Methods

This was a retrospective study with prospectively collected data. 1801 consecutive patients with *de novo* HCC were enrolled through our institution from 2011–2017.

### Results

There were 302 patients with stage 0, 783 with stage A, 242 with stage B, 358 with stage C, and 116 with stage D HCC. Treatment adhering to the modified BCLC guidelines recommendations was provided to 259 (85.8%) stage 0 patients, 606 (77.4%) stage A patients, 120 (49.6%) stage B patients, 93 (26.0%) stage C patients, and 83 (71.6%) stage D patients.

**Funding:** This study was supported by Grant CMRPG8J1281 from the Kaohsiung Chang Gung Memorial Hospital, Taiwan. Grant Recipient is Yi-Hao Yen. The funders had no role in study design, data collection and analysis, decision to publish, or preparation of the manuscript." There was no additional external funding received for this study.

**Competing interests:** he authors have declared that no competing interests exist.

## Conclusions

We reported treatment adhering to the modified BCLC guidelines at a high-volume liver surgery center in Taiwan. We found that non-adherence to the modified BCLC staging system was common in treating stage B and C patients.

## Introduction

Hepatocellular carcinoma (HCC) is one of the leading causes of cancer death worldwide [1, 2]. In patients with HCC tumors, unlike other solid tumors, the co-existence of two life-threatening diseases (i.e. cancer and liver cirrhosis) complicates the prognostic evaluation [3, 4]. The Barcelona Clinic Liver Cancer (BCLC) staging system includes tumor characteristics, liver function reserve (i.e. Child-Pugh class), and performance status; it is the most widely applied staging system for HCC and is recommended for treatment allocation and prognostic prediction [5].

In 2018, the European Association for the Study of Liver Diseases (EASL) proposed a modified BCLC staging system in which Child-Pugh A without any ascites is regarded as essential for all stages except stage D [6]. However, adherence to the modified guideline recommendations in the real world and the prognostic capability of the modified guidelines have yet to be determined.

Due to the heterogeneity of HCC and the multiple treatment modalities available, the management of HCC should be individualized rather than taking a "one size fits all" approach [7]. The EASL and the American Association for the Study of Liver Diseases (AASLD) guidelines both recommend that the situations of patients should be discussed by multidisciplinary teams (MDTs) in order to identify and tailor the most appropriate individualized treatment options [6, 7].

In the real world, adherence to the treatment algorithms for HCC recommended by the BCLC guidelines may be compromised by shortages of liver donors for liver transplants, the locations of tumors, and the presence of severe co-morbidities which unbalance the risk-benefit ratio of surgical treatments [8–10]. On the other hand, liver resection (LR) provides survival benefits across the BCLC stages [11], and the use of LR for HCCs outside of the BCLC guideline recommendations has been noted worldwide, including in Western centers [12].

Financial constraints play a pivotal role in adherence to the modified BCLC guideline recommendations. The current healthcare system in Taiwan, known as the National Health Insurance system, is totally different from those in Western countries. The system has covered more than 99% of Taiwan's citizens since 2004. Under the system, citizens are free to choose physicians and hospitals without referral, and all the treatment modalities for HCC are reimbursed. Furthermore, patients with cancers can apply for a catastrophic illness card. Therefore, patients with HCC do not have to pay anything when they receive medical care related to HCC. Nearly 90% of citizens are satisfied with the current health care system in Taiwan (https://www.mohw.gov.tw/cp-4251-50316-1.html).

Previous studies from Italy [13, 14] reported that non-adherence to the original BCLC guidelines [5] was common in real-world practice. Our primary aim in this study, relatedly, was to evaluate adherence to the modified BCLC guidelines and its impact on patient survival in a high-volume liver surgery center in East Asia, where there are higher etiological prevalence rates of hepatitis B virus (HBV) (including HBV–associated non-cirrhotic HCC) and

lower prevalence rates of hepatitis C virus (HCV) and non-alcoholic fatty liver disease (NAFLD) than in Western countries [7].

Furthermore, all of the current treatment modalities were available at the surgery center in question and were reimbursed by the National Health Insurance system of Taiwan. As such, certain variables that could modulate treatment decisions in clinical practice, including financial constraints, the availability of treatment procedures, and the expertise of the hospital, could be excluded.

Meanwhile, since the prognosis of a patient is affected by the treatment provided to that patient, our secondary aim in this study was to determine the prognostic capability of the modified BCLC staging system.

## Methods

### Patients

Data were extracted from the Kaohsiung Chang Gung Memorial Hospital HCC registry database, including the data for 1801 *de novo* HCC patients consecutively evaluated and managed at the hospital from January 2011 to December 2017. A flow chart of the patients' enrollment is shown in Fig 1. The data contained in the HCC registry database of the hospital were prospectively collected. The vital status of every single HCC patient was updated annually by linking to the website of the Ministry of Health and Welfare of Taiwan (https://hosplab.hpa.gov.tw/CSTIIS/index.aspx). The last update of vital statuses performed by linking to the website of the Ministry of Health and Welfare of Taiwan (https://hosplab.hpa.gov.tw/CSTIIS/index.aspx) in the current study was conducted in December 2018. The personnel who registered the HCC registry data checked the last follow-up date for each patient at 1, 3, and 5 years after the date of HCC diagnosis. The last follow-up date for each patient would be the last visit to our hospital, except for those patients who did not have follow-up visits at our hospital, who were contacted by phone. The last follow-up date was checked using this method until 5 years after the date of HCC diagnosis. If patients were still alive after December 2018 and more than 5 years since the date of HCC diagnosis, the last follow-up date would be 5 years since the date of HCC diagnosis. The method used to check the vital statuses and last follow-up dates of patients in the current study was different from performing manual medical record reviews. To avoid causing any confusion, we showed the 5-year overall survival (OS) rates in all figures in this study.

The following variables were recorded for each patient: age, gender, how HCC was diagnosed (clinically or pathologically), the etiology of chronic liver disease, serological parameters (i.e. total bilirubin level, transaminase level, albumin level, prothrombin time, and alpha fetoprotein (AFP) level), and Child-Pugh class [15]. Imaging study data (i.e. of contrast enhanced computed tomography (CT) or magnetic resonance imaging (MRI)) of the liver and chest X-ray data were also recorded. In those with severe renal impairment (i.e. an estimated glomerular filtration rate < 30 mL/min), non-enhanced MRI of the liver was performed as an alternative imaging study for staging. For patients who were surgical candidates, more comprehensive studies were performed (e.g. chest CT, thallium scan, etc.). The physician in charge evaluated the Eastern Cooperative Oncology Group (ECOG) performance status (PS) [5] and the presence or absence of hepatic encephalopathy or ascites for each patient. After complete disease staging (which was performed after collecting all of the aforementioned variables), each patient was classified according to the original BCLC staging system [5] and the 7th version the American Joint Committee on Cancer tumor-node-metastasis (AJCC TNM) staging system [16].

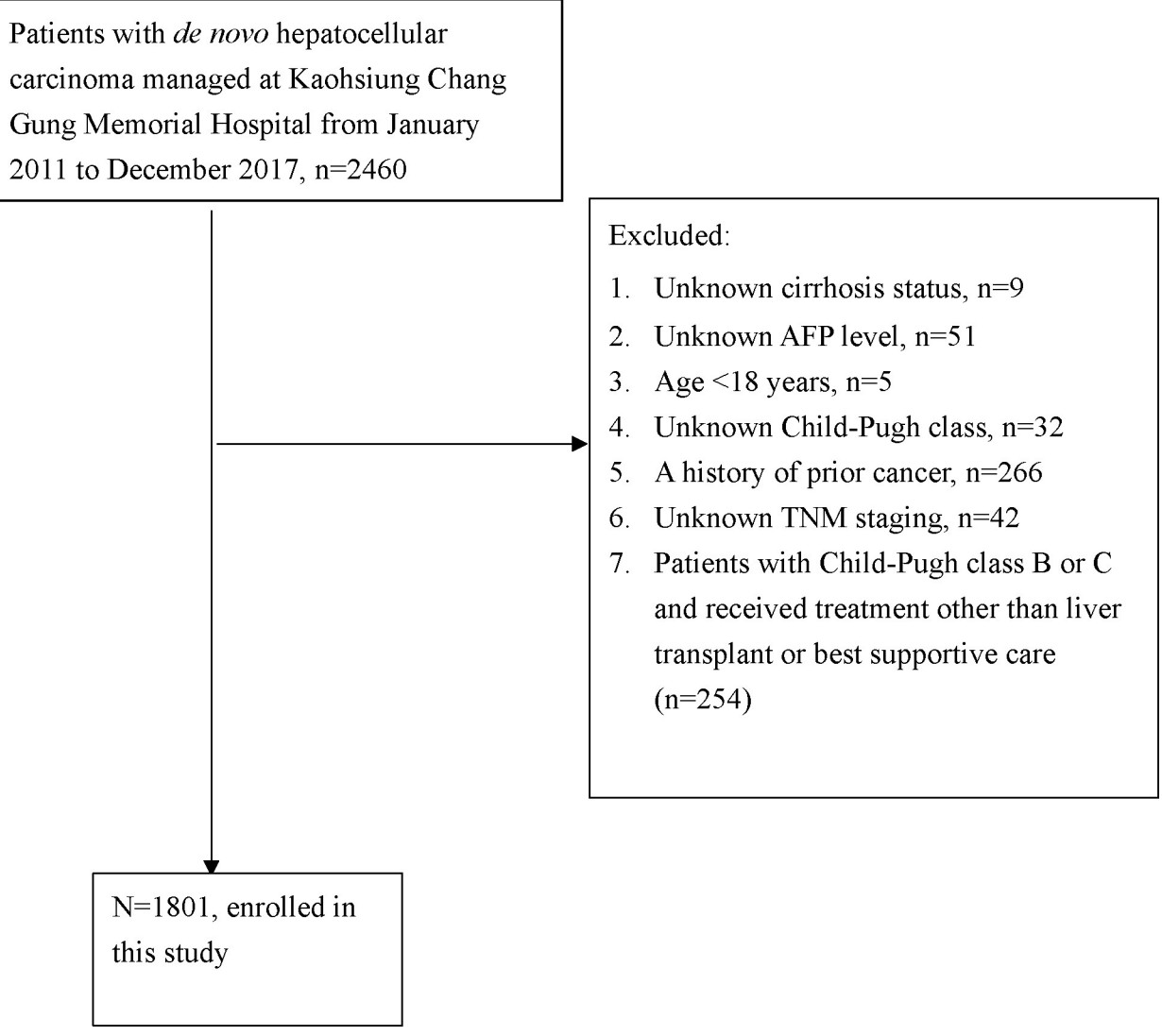

Patients with *de novo* hepatocellular carcinoma managed at Kaohsiung Chang Gung Memorial Hospital from January 2011 to December 2017, n=2460

Excluded:

1. Unknown cirrhosis status, n=9
2. Unknown AFP level, n=51
3. Age <18 years, n=5
4. Unknown Child-Pugh class, n=32
5. A history of prior cancer, n=266
6. Unknown TNM staging, n=42
7. Patients with Child-Pugh class B or C and received treatment other than liver transplant or best supportive care (n=254)

N=1801, enrolled in this study

**Fig 1. Flow chart of the patients enrolled.**

The staging for *de novo* HCC has to be completed and registered in the electronic medical records of the hospital in question by the physician in charge before treatment is provided. If the HCC staging is not completely registered, the computer system of the hospital will be locked down, such that the physician cannot order a treatment for the given HCC patient. This practice was initiated with the aim of effectively establishing the cancer registry. All hospitals with more than 50 beds in Taiwan have been required by the government to maintain a cancer registry since 2006.

The diagnosis of HCC was based on the assessment of a MDT and/or international guidelines [17–19]. Treatment options were evaluated according to the original BCLC staging system [5] and discussed by the MDT. The presence of cirrhosis was assessed by histology (i.e. an Ishak fibrosis score of 5 or 6) [20] or, if histological data was not available, was assessed using image studies. According to a previous study [21], having HCV (i.e. being anti-HCV-positive) was considered the primary etiology of liver disease for those who were anti-HCV-positive regardless of any other potential etiology. Meanwhile, having HBV (i.e. being hepatitis B

surface antigen (HBsAg)-positive) was considered the primary etiology of liver disease for any patients who were anti-HCV-negative. Alcohol use disorder was defined as habitual drinking.

All the procedures used in the study were in accordance with the ethical standards of the committees on human experimentation and with the Helsinki Declaration of 1975, as revised in 2008. The Institutional Review Board of Kaohsiung Chang Gung Memorial Hospital approved this study (reference number: 202000398B0) and waived the need for informed consent.

### Statistical analysis

The baseline characteristics and treatment modalities of the study cohort according to the modified BCLC stage were summarized as frequencies (percentages), means (standard deviations), or medians (ranges). The median overall survival (OS) rate of each BCLC stage with the 95% confidence interval (CI) was determined using Kaplan-Meier estimation. Comparisons of OS probability rates between groups were illustrated using Kaplan-Meier survival curves, and the survival differences between groups were estimated using the log-rank test. $P$ values less than 0.05 were considered statistically significant.

To evaluate the impact of non-adherence to the BCLC guideline recommendations on OS, the characteristics of patients with upward treatment stage migration and downward treatment stage migration according to the BCLC guideline recommendations were compared by chi-square test, Fisher's exact test, Mann-Whitney U test, Kurskal-Wallis test, one-way ANOVA, and independent-samples t test, as appropriate. The variables enrolled for univariate analysis were: age, sex, diagnosis method of HCC (pathology vs. clinical), tumor size, AJCC TNM stage, body mass index (BMI), etiology of liver disease, AFP level, cirrhosis status, Child-Pugh class, creatinine level, bilirubin level, and international normalized ratio (INR). The Cox model was adjusted for those variables with a p-value <0.05 or clinical relevance (e.g. AFP level). To avoid collinearity, redundant variables were not enrolled in the multivariate analysis. The reason why we enrolled the diagnosis method of HCC (pathology vs. clinical) was because if a patient's HCC was diagnosed clinically, it meant that the patient underwent non-surgical treatment. Furthermore, LR may improve survival across BCLC stages [11].

All statistical analyses were performed with Stata version 14.0. (StataCorp. 2015. Stata Statistical Software: Release 14. College Station, TX: StataCorp LP.).

## Results

### Baseline characteristics of the HCC population according to modified BCLC stage

The baseline characteristics of the 1801 HCC patients divided according to the modified BCLC staging system are summarized in Table 1. HCC was diagnosed by pathology in 1168 (64.9%) of the patients. The majority (73.5%) of the patients were males, and the median age of the patients was 62 years. Liver cirrhosis was present in 71.8% of the patients. HBV (45.1%) and then HCV (36.9%) were the most common etiological factors, and 90.8% of the patients had Child-Pugh class A liver disease. AFP > 200 ng/dL was noted in 27.3% of the patients. Tumor size > 5.0 cm was noted in 30.0% of the patients. 7th AJCC TNM stage 1 was noted in 53% of the patients. According to the modified BCLC staging system, there were 302 patients (16.8%) with stage 0, 783 (43.5%) with stage A, 242 (12.2%) with stage B, 358 (19.9%) with stage C, and 116 (6.4%) with stage D HCC. As expected, the proportions of patients with Child-Pugh class B or C were higher among the stage D patients, while the proportions of

**Table 1. Baseline characteristics of the HCC population, according to the modified BCLC stage.**

| | BCLC 0, n = 302 | BCLC A, n = 783 | BCLC B, n = 242 | BCLC C, n = 358 | BCLC D, n = 116 | All patients, n = 1801 |
|---|---|---|---|---|---|---|
| Age (years) median, range | 61 (31–87) | 63 (31–94) | 63.5 (29–87) | 62.5 (25–91) | 61 (31–96) | 62 (25–96) |
| Male | 198 (65.6%) | 572 (73.1%) | 187 (77.3%) | 284 (79.3%) | 82 (70.7%) | 1323 (73.5%) |
| BMI (kg/m$^2$), mean ±SD | 25.2 ± 3.8 | 25.4 ± 4 | 25.0 ± 3.8 | 24.2 ± 3.9 | 24.2 ± 3.9 | 25 ± 4.0 |
| Cirrhosis | 238 (78.8%) | 504 (64.4%) | 178 (73.6%) | 275 (76.8%) | 99 (85.3%) | 1294 (71.8%) |
| Child Pugh class | | | | | | |
| A | 302 (100%) | 767 (98.0%) | 236 (97.5%) | 310 (86.6%) | 20 (17.2%) | 1635 (90.8%) |
| B | (0%) | 16 (2.0%) | 6 (2.5%) | 45 (12.6%) | 24 (20.7%) | 91 (5.1%) |
| C | (0%) | (0%) | (0%) | 3 (0.8%) | 72 (62.1%) | 75 (4.2%) |
| Liver etiology | | | | | | |
| HBV | 134 (44.4%) | 351 (44.8%) | 110 (45.5%) | 165 (46.1%) | 52 (44.8%) | 812 (45.1%) |
| HCV | 137 (45.4%) | 305 (39.0%) | 83 (34.3%) | 104 (29.1%) | 36 (31%) | 665 (36.9%) |
| Alcohol use disorder | 7 (2.3%) | 15 (1.9%) | 10 (4.1%) | 15 (4.2%) | 6 (5.2%) | 53 (2.9%) |
| All negative | 24 (7.9%) | 112 (14.3%) | 39 (16.1%) | 74 (20.7%) | 22 (19.0%) | 271 (15.0%) |
| AFP (ng/dL) | | | | | | |
| <20 | 186 (61.6%) | 460 (58.7%) | 109 (45.0%) | 95 (26.5%) | 43 (37.1%) | 893 (49.6%) |
| 20–200 | 86 (28.5%) | 181 (23.1%) | 62 (25.6%) | 64 (17.9%) | 21 (18.1%) | 416 (23.1%) |
| >200 | 30 (9.9%) | 142 (18.1%) | 71 (29.3%) | 199 (55.6%) | 52 (44.8%) | 492 (27.3%) |
| Creatinine (mg/dL), median, range | 0.9 (0.4–14.1) | 1.0 (0.2–15.3) | 1.0 (0.2–9.8) | 1.0 (0.4–13.5) | 1.5 (0.4–20.2) | 1.0 (0.2–20.2) |
| Bilirubin (mg/dL), median, range | 1.0 (0.3–5.3) | 0.9 (0.2–17.3) | 1.1 (0.3–22.4) | 1.2 (0.1–43.0) | 4.6 (0.1–34.6) | 1.0 (0.1–43.0) |
| INR, median, range | 1.0 (0.9–2.5) | 1.0 (0.4–2.8) | 1.0 (0.9–2.6) | 1.1 (0.9–6.0) | 1.4 (0.9–6.0) | 1.0 (0.4–6.0) |
| Tumor size (cm)>5cm (%) | (0%) | 104 (13.3%) | 103 (42.6%) | 263 (73.5%) | 71 (61.2%) | 541 (30.0%) |
| HCC diagnosis | | | | | | |
| pathology | 177 (58.6%) | 571 (72.9%) | 163 (67.4%) | 210 (58.7%) | 47 (40.5%) | 1168 (64.9%) |
| Clinical | 125 (41.4%) | 212 (27.1%) | 79 (32.6%) | 148 (41.3%) | 69 (59.5%) | 633 (35.1%) |
| 7th AJCC TNM stage | | | | | | |
| 1 | 302 (100.0%) | 581 (74.2%) | 18 (7.4%) | 33 (9.2%) | 24 (20.7%) | 955 (53.0%) |
| 2 | (0%) | 202 (25.8%) | 117 (48.3%) | 24 (6.7%) | 17 (14.7%) | 360 (20.0%) |
| 3 | (0%) | (0%) | 107 (44.2%) | 195 (54.5%) | 43 (37.1%) | 348 (19.3%) |
| 4 | (0%) | (0%) | (0%) | 106 (29.6%) | 32 (27.6%) | 138 (7.7%) |

BMI, body mass index; INR, International Normalized Ratio; HBV, hepatitis B virus; HCV, hepatitis C virus; AJCC TNM, American Joint Committee on Cancer tumor-node-metastasis; HCC, Hepatocellular carcinoma; BCLC, Barcelona Clinic Liver Cancer.

patients with tumor size > 5cm, AFP > 200 ng/dL, or TNM stage 3 or 4 were higher among the stage C and D patients.

## Distribution of treatment received according to modified BCLC stage

Treatment adherent to the modified BCLC recommendations was provided to 259 (85.8%) of the stage 0 patients, 606 (77.4%) of the stage A patients, 120 (49.6%) of the stage B patients, 93 (26.0%) of the stage C patients, and 83 (71.6%) of the stage D patients. LR was one of the main treatment modalities for the stage 0 (29.8%), stage A (47%), stage B (34%), and stage C (17%) patients. Few patients in the overall cohort underwent liver transplant (n = 65 (3.6%)) (Table 2).

## Survival analysis

Over a median follow-up period of 17.3 (range = 0.2–84.7) months, 760 (42.2%) of the patients died. The causes of death were as follows: HCC-related death (n = 596), non-cancer-related

**Table 2. Distribution of patients and treatment options.** (N = 1801).

|  | BCLC 0, n = 302 | BCLC A, n = 783 | BCLC B, n = 242 | BCLC C, n = 358 | BCLC D, n = 116 |
|---|---|---|---|---|---|
| Resection | 90 (29.8%) | 368 (47%) | 82 (33.9%) | 61 (17.0%) | 2 (1.7%) |
| Transplant | 8 (2.6%) | 23 (2.9%) | 8 (3.3%) | 7 (2.0%) | 19 (16.4%) |
| RFA | 161 (53.3%) | 215 (27.5%) | 20 (8.3%) | 15 (4.2%) | 3 (2.6%) |
| TAE/TACE | 43 (14.2%) | 166 (21.2%) | 120 (49.6%) | 69 (19.3%) | 5 (4.3%) |
| Sorafenib | (0%) | (0%) | 2 (0.8%) | 93 (26.0%) | 1 (0.9%) |
| BSC | (0%) | 4 (0.5%) | 3 (1.2%) | 53 (14.8%) | 83 (71.6%) |
| Other | (0%) | 7 (0.9%) | 7 (2.9%) | 60 (16.8%) | 3 (2.6%) |

Other treatment (i.e. systemic chemotherapy, hepatic artery infusion chemotherapy or external beam radiation therapy). TACE/TAE, transcatheter arterial chemoembolization/embolization; BSC, best supportive care; RFA, radiofrequency ablation.

death (n = 138), unknown (n = 13), and non-HCC cancer-related death (n = 13). The median OS was 77.9 months (95% CI = 73-not available) among the stage 0 patients (Note: "not available" indicates that the 95% CI was not reached yet). It was 67.8 months (95% CI = 65.1–74.9) among the stage A patients, 35 months (95% CI = 25–47) among the stage B patients, 11 months (95% CI = 8.1–12) among the stage C patients, and 3.9 months (95% CI = 2.9–4.9) among the BCLC stage D patients (Fig 2). These results were compatible with the modified BCLC staging [5]. The 5-year OS rates were 71.1% for the stage 0, 59.9% for the stage A, 34.5% for the stage B, 16.2% for the stage C, and 20.4% for the stage D patients (Fig 2). The modified BCLC staging system showed a significant difference in the probability of survival across the different stages (Fig 2).

Among the stage 0 patients, the median OS was 81.1 months (95% CI = 63.9-not available) for those who received treatment according to the BCLC recommendations, and it was 73.0 months (95% CI = 55.1-not available) for those who received downward treatment stage migration (p = 0.73) (Fig 3A). Among the stage A patients, the median OS was 73.0 months (95% CI = 67.8–74.9) for those who received treatment according to the BCLC recommendations, and it was 45.8 months (95% CI = 36.0–57.1) for those who received downward treatment stage migration (p<0.001, Fig 3B). Among the stage B patients, the median survival was 81.1 months (95% CI = 44.8-not available) for those who received upward treatment stage migration, it was 28.2 months (95% CI = 23.0–36.0) for those who received treatment according to the BCLC recommendations, and it was 15.9 months (95% CI = 1.0–25.0) for those who received downward treatment stage migration (p<0.001, Fig 3C). Among the stage C patients, the median survival was 24.0 months (95% CI = 19.1–30.1) for those who received upward treatment stage migration, it was 6.1 months (95% CI = 4.9–8.1) for those who received treatment according to the BCLC recommendations, and it was 4.2 months (95% CI = 3.2–5.9) for those who received downward treatment stage migration (p<0.001, Fig 3D). Among the stage D patients, the median survival was not reached yet (95% CI = 40.9-not available) for those who received upward treatment stage migration, while it was 2.9 months (95% CI = 2.0–3.2) for those who received treatment according to the BCLC recommendations (p<0.001, Fig 3E).

## Characteristics of adherence and non-adherence to BCLC guideline recommendations in each BCLC stage

**BCLC stage 0.** Among BCLC stage 0 patients, there was a higher proportion of patients diagnosed by pathology, and the follow-up period was shorter in the patients treated according to the BCLC guideline recommendations than in the downward treatment stage migration

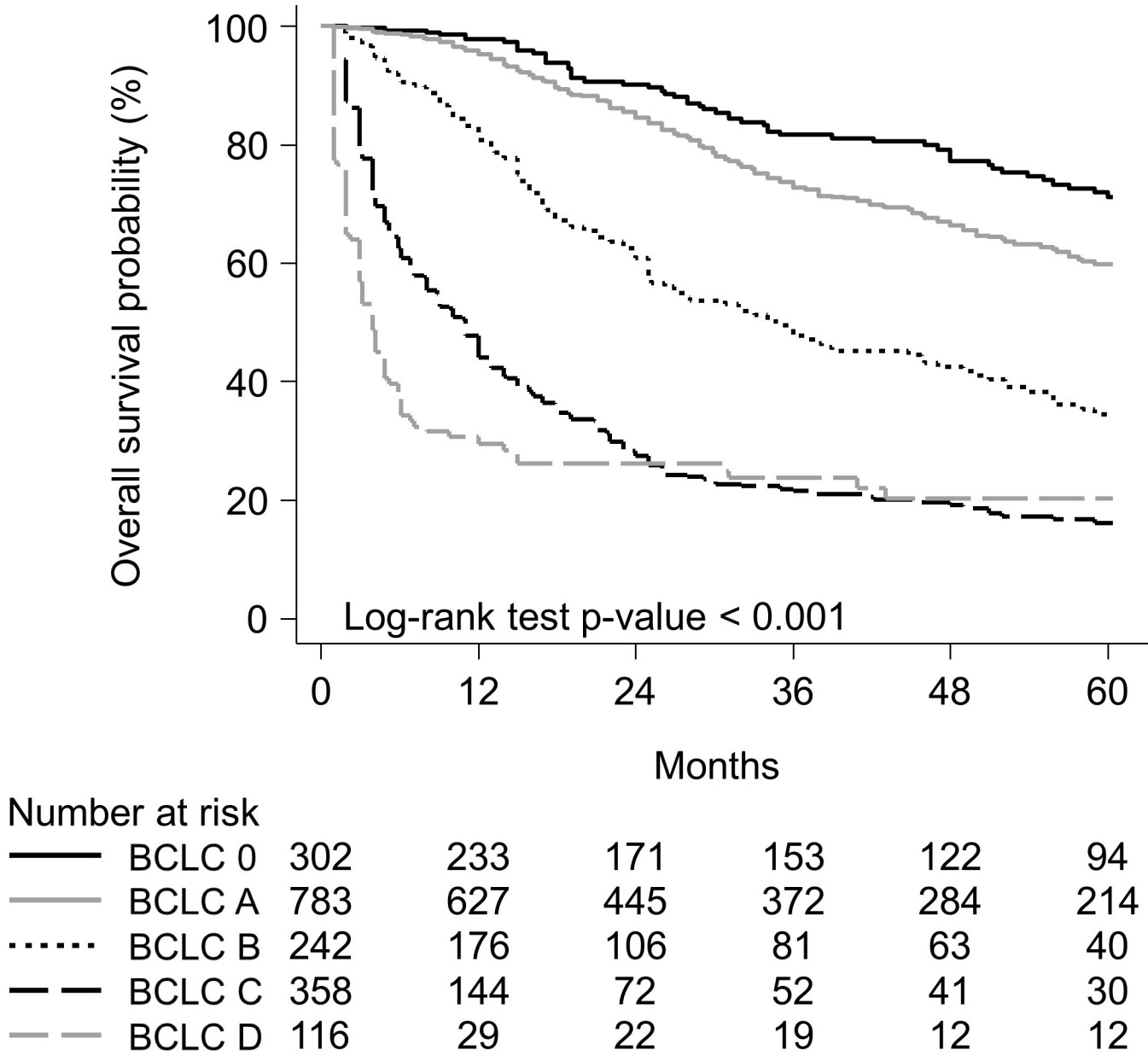

**Fig 2. Overall survival curves according to modified BCLC staging system.** BCLC 0 vs. A, p = 0.001; 0 vs. B, p < 0.001; 0 vs. C, p < 0.001; 0 vs. D, p < 0.001; A vs. B, p < 0.001; A vs. C, p < 0.001; A vs. D, p < 0.001; B vs. C, p < 0.001; B vs. D, p < 0.001; C vs. D, p = 0.015.

group. There were no significant difference between the two groups in terms of other characteristics (Table 3).

**BCLC stage A.** Among BCLC stage A patients, the mean age was lower, the proportion of patients with cirrhosis was lower, the bilirubin and INR levels were lower, and the proportion of HCCs diagnosed by pathology and with TNM stage 1 was higher in the patients treated according to the BCLC guideline recommendations than in the downward treatment stage migration group. There were no significant differences between the two groups in terms of other characteristics (Table 4).

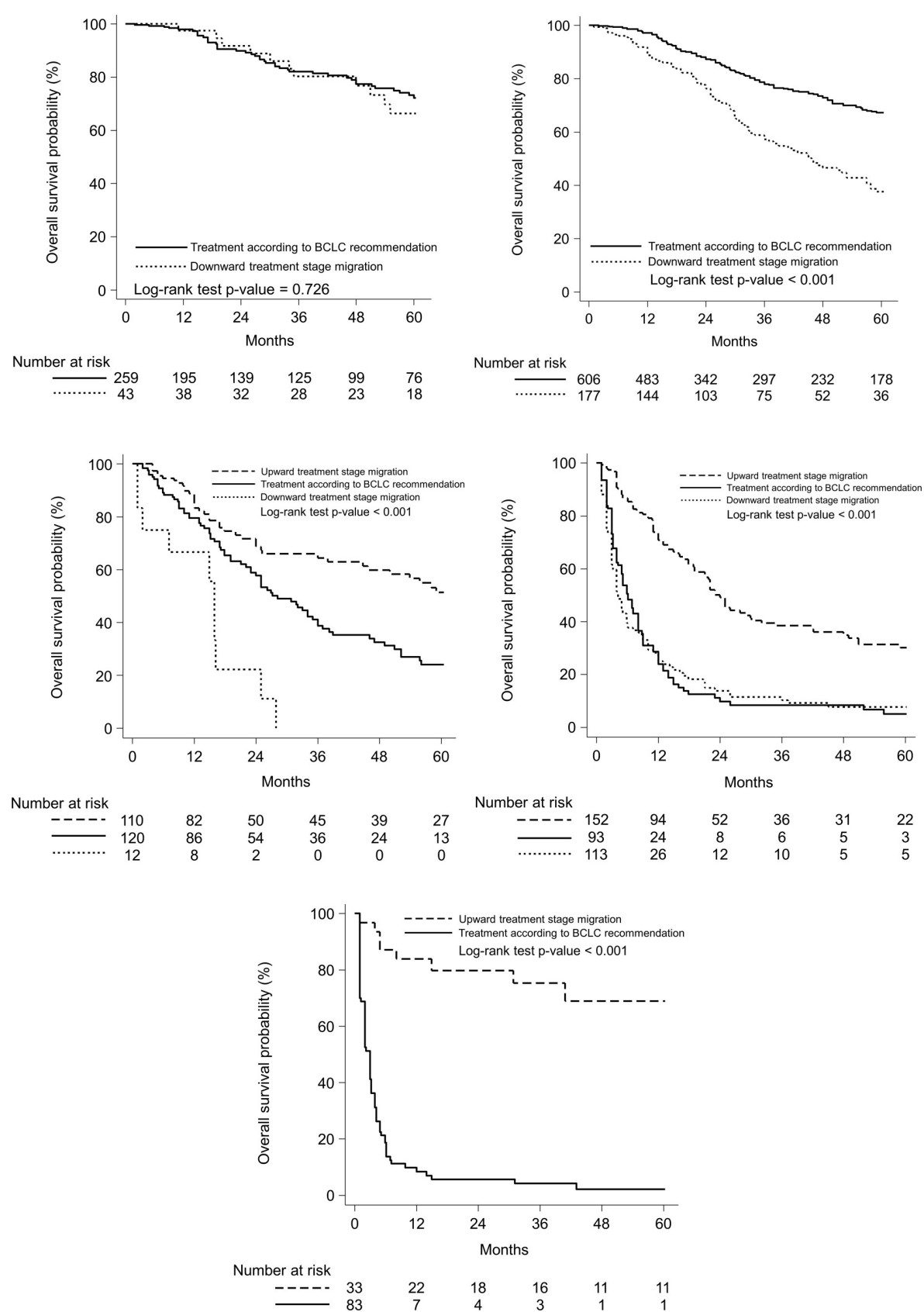

**Fig 3.** A. Overall survival curves in BCLC 0 patients, according to received treatment. Solid line = Treatment according to BCLC recommendation, median survival 81.1 months (95% CI = 63.9- not available). Dotted line = Downward treatment stage migration. median survival 73.0 months (95%- CI = 55.1-not available). B. Overall survival curves in modified BCLC A patients, according to received treatment. Solid line = Treatment according to BCLC recommendation, median survival 73.0 months (95% CI = 67.8–74.9). Dotted line = Downward treatment stage migration. median survival 45.8 months (95% C I = 36.0–57.1). C. Overall survival curves in modified BCLC B patients, according to received treatment. Dashed line = upward treatment stage migration, median survival 81.1 months (95% CI = 44.8- not available). Solid line = Treatment according to BCLC recommendation, median survival 28.2 months (95% CI = 23.0–36.0). Dotted line = downward treatment stage migration, median survival 15.9 months (95% CI = 1.0–25.0). Upward treatment vs. recommended, p = 0.001; Upward treatment vs. downward, p < 0.001; recommended treatment vs. downward, p = 0.001. D. Overall survival curves in modified BCLC C patients, according to received treatment. Dashed line = upward treatment stage migration, median survival 24.0 months (95% CI = 19.1–30.1). Solid line = Treatment according to BCLC recommendation, median survival 6.1 months (95% CI = 4.9–8.1). Dotted line = downward treatment stage migration, median survival 4.2 months (95% CI = 3.2–5.9). Upward treatment vs. recommended, p < 0.001; Upward treatment vs. downward, p < 0.001; recommended treatment vs. downward, p = 0.812. E. Overall survival curves in modified BCLC D patients, according to received treatment. Dashed line = upward treatment stage migration, median survival not reached (95% CI = 40.9-not available). Solid line = Treatment according to BCLC recommendation, median survival 2.9 months (95% CI = 2.0–3.2).

**Table 3. Characteristics of patients with adherence and non-adherence to BCLC guidelines recommendation in BCLC stage 0.**

| | Treatment according to the BCLC guidelines recommendations, n = 259 | Downward treatment stage migration, n = 43 | p |
|---|---|---|---|
| Age (years), median (range) | 61(31–87) | 62(42–85) | 0.54 |
| Male, n (%) | 168 (64.9%) | 30 (69.8%) | 0.531 |
| BMI (kg/m$^2$), mean ±SD | 25.0±3.5 | 26.0±5.1 | 0.131 |
| Cirrhosis, n (%) | 200 (77.2%) | 38 (88.4%) | 0.097 |
| Child Pugh class, n (%) | | | |
| A | 259 (100%) | 43(100%) | |
| B | | | |
| C | | | |
| Etiology of chronic liver disease | | | 0.137 |
| HBV, n (%) | 114 (44.0%) | 20 (46.5%) | |
| HCV, n (%) | 122 (47.1%) | 15 (34.9%) | |
| Alcohol, n (%) | 5 (1.9%) | 2 (4.7%) | |
| All negative, n (%) | 18 (6.9%) | 6 (14.0%) | |
| AFP (ng/ml), n (%) | | | 0.130 |
| <20 | 165 (63.7%) | 21 (48.8%) | |
| 20–200 | 71 (27.4%) | 15 (34.9%) | |
| >200 | 23 (8.9%) | 7 (16.3%) | |
| Creatinine (mg/dL), median (range) | 0.9(0.4–14.1) | 0.9(0.5–3.3) | 0.584 |
| Bilirubin (mg/dL), median (range) | 1.0(0.3–5.0) | 1.0(0.5–5.3) | 0.364 |
| INR, median (range) | 1.0(0.9–2.5) | 1.0(0.9–1.6) | 0.081 |
| HCC diagnosis, n (%) | | | <0.001 |
| Pathology | 165(63.7%) | 12(27.9%) | |
| Clinical | 94(36.3%) | 31 (72.1%) | |
| 7$^{th}$ AJCC TNM stage, n (%) | | | 1.00 |
| 1 | 256(98.8%) | 43(100%) | |
| 2 | 3(1.2%) | | |
| 3 | | | |
| 4 | | | |
| Follow up times (months), median (IQR) | 32.4 (11.6–60.3) | 51.1 (20.4–63.2) | 0.004 |

BMI, body mass index; INR, International Normalized Ratio; HBV, hepatitis B virus; HCV, hepatitis C virus; AJCC TNM, American Joint Committee on Cancer tumor-node-metastasis; HCC, Hepatocellular carcinoma; BCLC, Barcelona Clinic Liver Cancer; IQR, interquartile range.

**Table 4. Characteristics of patients with adherence and non-adherence to BCLC guidelines recommendation in BCLC stage A.**

| | Treatment according to the BCLC guidelines recommendation. n = 606 | Downward treatment stage migration, n = 177 | p |
|---|---|---|---|
| Age (years), median (range) | 63.0(31–88) | 63.0(35–94) | 0.018 |
| Male, n (%) | 445 (73.4%) | 127(71.8%) | 0.657 |
| BMI (kg/m$^2$), mean ±SD | 25.3±4.0 | 25.8±4.1 | 0.182 |
| Cirrhosis, n (%) | 343 (56.6%) | 161(91.0%) | <0.001 |
| Child Pugh class, n (%) | | | 0.545 |
| A | 592(97.7%) | 175(98.9%) | |
| B | 14(2.3%) | 2(1.1%) | |
| C | 0 | 0 | |
| Etiology of chronic liver disease | | | 0.460 |
| HBV | 277(45.7%) | 74(41.8%) | |
| HCV | 237(39.1%) | 68(38.4%) | |
| Alcohol | 10(1.7%) | 5(2.8%) | |
| All negative | 82(13.5%) | 30(16.9%) | |
| AFP (ng/ml), n (%) | | | 0.124 |
| <20 | 366(60.4%) | 94(53.1%) | |
| 20–200 | 140(23.1%) | 43(24.3%) | |
| >200 | 100(16.5%) | 40(22.6%) | |
| Creatinine (mg/dL), median (range) | 1.0(0.2–15.3) | 1.0(0.4–15.0) | 0.919 |
| Bilirubin (mg/dL), median (range) | 0.9(0.2–7.6) | 1.1(0.2–17.3) | <0.001 |
| INR, median (range) | 1.0(0.4–2.8) | 1.0(0.9–1.6) | <0.001 |
| Tumor size>5.0cm, n (%) | 82(13.6%) | 22(12.6%) | 0.744 |
| HCC diagnosis, n (%) | | | <0.001 |
| Pathology | 511(84.3%) | 60(33.9%) | |
| Clinical | 95(15.7%) | 117(66.1%) | |
| 7$^{th}$ AJCC TNM stage, n (%) | | | <0.001 |
| 1 | 482(79.5%) | 99(55.9%) | |
| 2 | 122(20.1%) | 77(43.5%) | |
| 3 | 2(0.3%) | 1 (0.6%) | |
| 4 | | | |
| Follow up times (months), median (IQR) | 34.8 (12.3–60.4) | 29.9 (12.6–53.6) | 0.233 |

BMI, body mass index; INR, International Normalized Ratio; HBV, hepatitis B virus; HCV, hepatitis C virus; AJCC TNM, American Joint Committee on Cancer tumor-node-metastasis; HCC, Hepatocellular carcinoma; BCLC, Barcelona Clinic Liver Cancer; IQR, interquartile range.

**BCLC stage B.** Among BCLC stage B patients, the proportion of patients with cirrhosis was lowest, the bilirubin and INR levels were lowest, and the proportion of HCCs diagnosed by pathology and with TNM stage 1 was highest in the upward treatment migration group compared with the other groups. Meanwhile, the proportion of patients with Child-Pugh class A was highest and the follow-up period was longer in the patients treated according to the BCLC guideline recommendations than in the other patient groups. There were no significant differences in terms of other characteristics among the groups (Table 5).

**BCLC stage C.** Among BCLC stage C patients, the mean age was highest, the proportions of male patients and those with tumor size >5.0 cm were lowest, the proportion of patients with AFP<20 ng/ml was highest, the proportion of HCCs diagnosed by pathology and with TNM stage 1 was highest, the bilirubin and INR levels were lowest, and the follow-up period was longer in the upward treatment stage migration group than in the other patient groups.

**Table 5. Characteristics of patients with adherence and non-adherence to BCLC guidelines recommendation in BCLC stage B.**

| | Treatment according to the BCLC guidelines recommendation, n = 120 | Upward treatment stage migration, n = 110 | Downward treatment stage migration, n = 12 | p |
|---|---|---|---|---|
| Age (years) median (range) | 64(37–87) | 61.5(29–83) | 66(32–85) | 0.097 |
| Male, n (%) | 92(76.7%) | 88(80.0%) | 7 (58.3) | 0.230 |
| BMI (kg/m$^2$), mean ±SD | 25.4 ±4.0 | 24.7 ±3.7 | 24.6 ±3.5 | 0.36 |
| Cirrhosis, n (%) | 111(92.5%) | 59(53.6%) | 8(66.7%) | <0.001 |
| Child Pugh class, n (%) | | | | <0.001 |
| A | 120(100%) | 107(97.3%) | 9(75.0%) | |
| B | 0 | 3(2.7%) | 3(25.3%) | |
| C | 0 | 0 | 0 | |
| Etiology of chronic liver disease, n(%) | | | | 0.471 |
| HBV | 53(44.2%) | 53(48.2%) | 4(33.3%) | |
| HCV | 44(36.7%) | 36(32.7%) | 3(25.0%) | |
| Alcohol | 6(5.0%) | 3(2.7%) | 1(8.3%) | |
| All negative | 17(14.2%) | 18(16.4%) | 4(33.3%) | |
| AFP (ng/ml), n (%) | | | | 0.289 |
| <20 | 55(45.8%) | 50(45.5%) | 4(33.3%) | |
| 20–200 | 33(27.5) | 28(25.5%) | 1(8.3%) | |
| >200 | 32(26.7%) | 32(29.1%) | 7(58.3%) | |
| Creatinine (mg/dL), median (range) | 1.0(0.6–9.8) | 1.0(0.5–9.4) | 1.0(0.2–3.1) | 0.604 |
| Bilirubin (mg/dL), median (range) | 1.0(0.2–16.2) | 0.9(0.3–9.3) | 1.15(0.5–22.4) | 0.023 |
| INR, median (range) | 1.0(0.9–2.6) | 1.0(0.9–1.7) | 1.1(1.0–2.2) | 0.001 |
| Tumor size>5.0cm, n (%) | 52(43.7%) | 44(40.0%) | 7(58.3%) | 0.455 |
| HCC diagnosis, n (%) | | | | <0.001 |
| pathology | 56(46.7%) | 103(93.6%) | 4(33.3%) | |
| Clinical | 64(53.3%) | 7(6.4%) | 8(66.7%) | |
| 7[th] AJCC TNM stage, n (%) | | | | <0.001 |
| 1 | 0 | 18(16.4%) | 0 | |
| 2 | 68(56.7%) | 45(40.9%) | 4(33.3%) | |
| 3 | 52(43.3%) | 47(42.7%) | 8(66.7%) | |
| 4 | 0 | 0 | 0 | |
| Follow up times (months), median (IQR) | 19.0 (10.6–39.6) | 17.8 (11.3–58.9) | 14.1 (3.0–16.2) | 0.043 |

BMI, body mass index; INR, International Normalized Ratio; HBV, hepatitis B virus; HCV, hepatitis C virus; AJCC TNM, American Joint Committee on Cancer tumor-node-metastasis; HCC, Hepatocellular carcinoma; BCLC, Barcelona Clinic Liver Cancer; IQR, interquartile range.

The proportion of patients with Child-Pugh class A was highest in the patients treated according to the BCLC guideline recommendations compared with the other patient groups. There were no significant difference in terms of other characteristics among the groups (Table 6).

**BCLC stage D.** Among BCLC stage D patients, the mean age was younger, the BMI was higher, the proportion of patients with Child-Pugh class A was higher, the proportion of patients with AFP< 20ng/ml was higher, the proportion of HCCs diagnosed by pathology and with TNM stage 1 was higher, the creatinine and bilirubin levels were lower, the proportion of patients with tumor size > 5.0 cm was lower and the follow-up period was longer in the upward treatment stage migration group than in the patients treated according to the BCLC

**Table 6. Characteristics of patients with adherence and non-adherence to BCLC guidelines recommendation in BCLC stage C.**

| | Treatment according to the BCLC guidelines recommendation, n = 93 | Upward treatment stage migration, n = 152 | Downward treatment stage migration, n = 113 | p |
|---|---|---|---|---|
| Age (years), median (range) | 59(33–87) | 64(30–91) | 63(25–87) | 0.021 |
| Male, n (%) | 82(88.2%) | 105(69.1%) | 97(85.8%) | <0.001 |
| BMI (kg/m$^2$), mean ±SD | 23.8±3.6 | 24.5±3.9 | 24.2±4.2 | 0.509 |
| Cirrhosis, n (%) | 73 (78.5%) | 110(72.4%) | 92(81.4%) | 0.204 |
| Child Pugh class, n (%) | | | | <0.001 |
| A | 93(100%) | 147(96.7%) | 70(61.9%) | |
| B | 0 | 5(3.3%) | 40(35.4%) | |
| C | 0 | 0 | 3(2.7%) | |
| Etiology of chronic liver disease, n(%) | | | | 0.117 |
| HBV | 42(45.2%) | 72(47.4%) | 51(45.1%) | |
| HCV | 27(29.0%) | 49(32.2%) | 28(24.8%) | |
| Alcohol | 5(5.4%) | 1(0.7%) | 9(8.0%) | |
| All negative | 19(20.4%) | 30(19.7%) | 25(22.1%) | |
| AFP (ng/ml), n(%) | | | | <0.001 |
| <20 | 12(12.9%) | 61(40.1%) | 22(19.5%) | |
| 20–200 | 15(16.1%) | 26(17.1%) | 23(20.4%) | |
| >200 | 66(71.0%) | 65(42.8%) | 68(60.2%) | |
| Creatinine (mg/dL), median (range) | 1.0(0.4–3.4) | 1.0(0.5–13.5) | 1.1(0.4–11.7) | 0.058 |
| Bilirubin (mg/dL), median (range) | 1.2(0.4–15.3) | 1.0(0.1–6.2) | 1.9(0.4–43.0) | <0.001 |
| INR, median (range) | 1.1(0.9–5.0) | 1.0(0.9–2.2) | 1.1(0.9–6.0) | <0.001 |
| Tumor size >5cm, n (%) | 84(94.4%) | 91(60.3%) | 88(79.3%) | <0.001 |
| HCC diagnosis, n (%) | | | | <0.001 |
| pathology | 51(54.8%) | 107(70.4%) | 52(46.0) | |
| Clinical | 42(45.2%) | 45(29.6%) | 61(54.0%) | |
| 7$^{th}$ AJCC TNM stage, n (%) | | | | <0.001 |
| 1 | 0 | 30 (19.7%) | 3(2.7%) | |
| 2 | 0 | 22(14.5%) | 2(1.8%) | |
| 3 | 49(52.7%) | 82(53.9%) | 64(56.6%) | |
| 4 | 44(47.3%) | 18(11.8%) | 44(38.9%) | |
| Follow up times (months), median (IQR) | 6.0 (3.0–11.9) | 14.0 (9.1–35.0) | 4.3 (2.3–10.9) | <0.001 |

BMI, body mass index; INR, International Normalized Ratio; HBV, hepatitis B virus; HCV, hepatitis C virus; AJCC TNM, American Joint Committee on Cancer tumor-node-metastasis; HCC, Hepatocellular carcinoma; BCLC, Barcelona Clinic Liver Cancer; IQR, interquartile range.

guideline recommendations. There were no significant difference between the groups in terms of other characteristics (Table 7).

## Predictors of OS by univariate and multivariate analysis in each BCLC stage

Among BCLC stage 0 patients, INR (HR = 5.189, 95%CI = 1.933–13.933, p = 0.001) was the only independent factor associated with OS in the multivariate analysis, while treatment adherence vs. non-adherence was not (Table 8).

**Table 7. Characteristics of patients with adherence and non-adherence to BCLC guidelines recommendation in BCLC stage D.**

| | Treatment according to the BCLC guidelines recommendation, n = 83 | Upward treatment stage migration, n = 33 | p |
|---|---|---|---|
| Age (years), median (range) | 63(31–96) | 58(37–83) | 0.039 |
| Male, n (%) | 57(68.7%) | 25(75.8%) | 0.45 |
| BMI (kg/m$^2$), mean ±SD | 23.7±3.8 | 26.3±5.5 | 0.021 |
| Cirrhosis, n (%) | 74(89.2%) | 25(75.8%) | 0.083 |
| Child Pugh class, n (%) | | | <0.001 |
| A | 6(7.2%) | 14(42.4%) | |
| B | 22(26.5%) | 2(6.1%) | |
| C | 55(66.3%) | 17(51.5%) | |
| Etiology of chronic liver disease, n (%) | | | 0.266 |
| HBV | 41(49.4%) | 11(33.3%) | |
| HCV | 23(27.7%) | 13(39.4%) | |
| Alcohol | 3(3.6%) | 3(9.1%) | |
| All negative | 16(19.3%) | 6(18.2) | |
| AFP (ng/ml), n (%) | | | <0.001 |
| <20 | 22(26.5%) | 21(63.6%) | |
| 20–200 | 15(18.1%) | 6(18.2%) | |
| >200 | 46(55.4%) | 6(18.2%) | |
| Creatinine (mg/dL), median (range) | 1.8(0.4–11.3) | 0.9(0.5–202) | <0.001 |
| Bilirubin (mg/dL), median (range) | 7.7(0.1–34.6) | 2.9(0.5–33.3) | <0.001 |
| INR, median (range) | 1.4(0.9–6.0) | 1.3(1.0–1.9) | 0.056 |
| Tumor size >5.0cm, n (%) | 62(76.5%) | 9(27.3%) | <0.001 |
| HCC diagnosis, n (%) | | | <0.001 |
| pathology | 19(22.9%) | 28(84.8%) | |
| Clinical | 64(77.1%) | 5(15.2%) | |
| 7th AJCC TNM stage, n (%) | | | <0.001 |
| 1 | 7(8.4%) | 17(51.5%) | |
| 2 | 9(10.8%) | 8(24.2%) | |
| 3 | 40(48.2%) | 3(9.1%) | |
| 4 | 27(32.5%) | 5(15.2%) | |
| Follow up times (months), median (IQR) | 2.2 (1.2–4.7) | 30.9 (8.8–60.1) | <0.001 |

BMI, body mass index; INR, International Normalized Ratio; HBV, hepatitis B virus; HCV, hepatitis C virus; AJCC TNM, American Joint Committee on Cancer tumor-node-metastasis; HCC, Hepatocellular carcinoma; BCLC, Barcelona Clinic Liver Cancer; IQR, interquartile range.

**Table 8. Univariate and multivariate analysis of overall survival in BCLC stage 0.**

| Variables | Univariate | | | multivariate | | |
|---|---|---|---|---|---|---|
| | HR | 95%CI | p | HR | 95%CI | p |
| Downward treatment stage migration vs. Adherence to BCLC guidelines recommendation | 1.117 | 0.587–2.213 | 0.736 | 1.007 | 0.525–1.930 | 0.984 |
| Cirrhosis (yes vs. no) | 2.320 | 0.924–5.824 | 0.073 | 1.854 | 0.727–4.731 | 0.196 |
| AFP ng/ml, >200 vs. ≦200 | 0.439 | 0.137–1.409 | 0.166 | 0.505 | 0.156–1.637 | 0.255 |
| INR (per 1 increase) | 6.055 | 2.334–15.704 | <0.001 | 5.189 | 1.933–13.933 | 0.001 |

INR, International Normalized Ratio; BCLC, Barcelona Clinic Liver Cancer; AFP, alpha-feto protein.

**Table 9. Univariate and multivariate analysis of overall survival in BCLC stage A.**

| Variables | Univariate | | | Multivariate | | |
|---|---|---|---|---|---|---|
| | HR | 95%CI | p | HR | 95%CI | p |
| Downward treatment stage migration vs. adherence to BCLC guidelines recommendation | 2.433 | 1.850–3.200 | <0.001 | 1.881 | 1.342–2.638 | <0.001 |
| Age (years) >65 vs. ≤65 | 1.441 | 1.104–1.881 | 0.007 | 1.369 | 1.041–1.801 | 0.025 |
| Cirrhosis (yes vs. no) | 2.029 | 1.475–2.790 | <0.001 | 1.539 | 1.081–2.193 | 0.017 |
| AFP ng/ml, >200 vs. ≦200 | 1.893 | 1.397–2.566 | <0.001 | 1.804 | 1.321–2.464 | <0.001 |
| Bilirubin (mg/dL) (per 1 unit increase) | 1.179 | 1.058–1.314 | 0.003 | 1.094 | 0.922–1.297 | 0.303 |
| INR (per 1 increase) | 1.751 | 0.882–3.476 | 0.109 | 1.364 | 0.538–3.453 | 0.513 |
| HCC diagnosis (Pathology vs. clinical) | 0.589 | 0.448–0.775 | <0.001 | 0.980 | 0.700–1.373 | 0.907 |
| 7th AJCC TNM stage, 2 or 3 vs. 1 | 1.315 | 0.990–1.748 | 0.059 | 1.108 | 0.822–1.493 | 0.502 |

INR, International Normalized Ratio; AJCC TNM, American Joint Committee on Cancer tumor-node-metastasis; BCLC, Barcelona Clinic Liver Cancer; AFP, alpha-feto protein; HCC, Hepatocellular carcinoma.

Among BCLC stage A patients, downward treatment stage migration (HR = 1.881, 95% CI = 1.342–2.638, p<0.001), age > 65 years (HR = 1.369, 95%CI = 1.041–1.801, p = 0.025), cirrhosis status (HR = 1.539, 95%CI = 1.081–2.193, p = 0.017), and AFP level > vs. ≦200 ng/ml (HR = 1.804, 95%CI = 1.321–2.464, p<0.001) were independent predictors for OS (Table 9).

Among BCLC stage B patients, upward treatment stage migration (HR = 0.413, 95% CI = 0.246–0.693, p = 0.001), Child-Pugh class B vs. A (HR = 4.656, 95%CI = 1.343–16.145, p = 0.015), AFP level > vs. ≦200 ng/ml (HR = 2.799, 95%CI = 1.891–4.142, p<0.001), and TNM stage 3 vs. 1 or 2 (HR = 2.865, 95%CI = 1.905–4.308, p<0.001) were independent predictors for OS (Table 10).

Among BCLC stage C patients, upward treatment stage migration (HR = 0.464, 95% CI = 0.324–0.663, p<0.001), Child-Pugh class B or C vs. A (HR = 3.096, 95%CI = 2.060–4.654, p<0.001), HCC diagnosed by pathology (HR = 0.659, 95%CI = 0.505–0.860, p = 0.002), and TNM stage 4 vs. 1 (HR = 2.731, 95%CI = 1.460–5.110, p = 0.002) were independent predictors for OS (Table 11).

Among BCLC stage D patients, upward treatment stage migration (HR = 0.144, 95% CI = 0.043–0.483, p = 0.002), BMI per 1 unit increase (HR = 1.077, 95%CI = 1.003–1.157, p = 0.004), AFP level > vs. ≦200 ng/ml (HR = 2.246, 95%CI = 1.233–4.093, p = 0.008), HCC diagnosed by pathology vs. clinical (HR = 0.466, 95%CI = 0.239–0.909, p = 0.025), and TNM

**Table 10. Univariate and multivariate analysis of overall survival in BCLC stage B.**

| Variables | Univariate | | | Multivariate | | |
|---|---|---|---|---|---|---|
| | HR | 95%CI | p | HR | 95%CI | p |
| Upward treatment stage migration vs. adherence to BCLC guidelines recommendation | 0.510 | 0.345–0.754 | 0.001 | 0.413 | 0.246–0.693 | 0.001 |
| Age (years) >65 vs. ≤65 | 1.302 | 0.908–1.869 | 0.152 | 1.229 | 0.844–1.791 | 0.283 |
| Cirrhosis (Yes vs. no) | 1.043 | 0.694–1.568 | 0.840 | 1.063 | 0.619–1.826 | 0.824 |
| Child Pugh class (B vs. A) | 1.283 | 0.407–4.044 | 0.670 | 4.656 | 1.343–16.145 | 0.015 |
| AFP ng/ml, >200 vs. < = 200 | 2.967 | 2.051–4.290 | <0.001 | 2.799 | 1.891–4.142 | <0.001 |
| HCC diagnosis (Pathology vs. clinical) | 0.719 | 0.498–1.038 | 0.078 | 1.033 | 0.666–1.603 | 0.883 |
| 7th AJCC TNM stage | | | | | | |
| 3 vs. 1/2 | 2.524 | 1.752–3.634 | <0.001 | 2.865 | 1.905–4.308 | <0.001 |

AJCC TNM, American Joint Committee on Cancer tumor-node-metastasis; BCLC, Barcelona Clinic Liver Cancer; AFP, alpha-feto protein; HCC, Hepatocellular carcinoma.

**Table 11.** Univariate and multivariate analysis of overall survival in BCLC stage C.

| Variables | Univariate | | | Multivariate | | |
|---|---|---|---|---|---|---|
| | HR | 95%CI | p | HR | 95%CI | p |
| Downward treatment stage migration vs. adherence to BCLC guidelines recommendation | 1.054 | 0.787–1.409 | 0.725 | 0.895 | 0.634–1.264 | 0.530 |
| Upward treatment stage migration vs. adherence to BCLC guidelines recommendation | 0.324 | 0.239–0.440 | <0.001 | 0.464 | 0.324–0.663 | <0.001 |
| Age (years) >65 vs. ≤65 | 0.810 | 0.632–1.038 | 0.096 | 0.940 | 0.721–1.226 | 0.648 |
| Male vs. female | 1.145 | 0.848–1.546 | 0.378 | 0.864 | 0.620–1.204 | 0.387 |
| Child Pugh class (B or C vs. A) | 3.496 | 2.525–4.841 | <0.001 | 3.096 | 2.060–4.654 | <0.001 |
| AFP ng/ml, >200 vs. ≦200 | 1.528 | 1.197–1.952 | 0.001 | 1.129 | 0.856–1.488 | 0.390 |
| Tumor size >5cm (yes vs. no) | 1.620 | 1.209–2.171 | 0.001 | 1.110 | 0.788–1.563 | 0.551 |
| HCC diagnosis (Pathology vs. clinical) | 0.492 | 0.387–0.627 | <0.001 | 0.659 | 0.505–0.860 | 0.002 |
| 7th AJCC TNM stage | | | | | | |
| 2 vs. 1 | 0.599 | 0.269–1.335 | 0.210 | 0.808 | 0.353–1.849 | 0.613 |
| 3 vs. 1 | 1.659 | 1.016–2.710 | 0.043 | 1.355 | 0.754–2.432 | 0.309 |
| 4 vs. 1 | 4.116 | 2.480–6.381 | <0.001 | 2.731 | 1.460–5.110 | 0.002 |

AJCC TNM, American Joint Committee on Cancer tumor-node-metastasis; BCLC, Barcelona Clinic Liver Cancer; AFP, alpha-feto protein; HCC, Hepatocellular carcinoma.

stage 4 vs. 1 (HR = 4.874, 95%CI = 1.462–15.651, p = 0.01) were independent predictors for OS (Table 12).

## Discussion

Child-Pugh classification is the most widely applied classification in assessing liver function reserve [15]. The EASL guidelines recommend the following: Preserved liver function should refer to Child-Pugh class A without any ascites, which are instead seen primarily in patients with end-stage or "decompensated" liver function [6]. The limitations of Child-Pugh classification include that hyperbilirubinemia may be associated with a non-liver-related disease (e.g. hemolysis) and that the albumin level may be affected by a non-liver-related disease (e.g.

**Table 12.** Univariate and multivariate analysis of overall survival in BCLC stage D.

| Variables | Univariate | | | Multivariate | | |
|---|---|---|---|---|---|---|
| | HR | 95%CI | p | HR | 95%CI | p |
| Upward treatment migration vs. adherence to BCLC guidelines recommendation | 0.114 | 0.057–0.225 | <0.001 | 0.144 | 0.043–0.483 | 0.002 |
| Age (years) >65 vs. ≤65 | 1.951 | 1.272–2.992 | 0.002 | 1.402 | 0.794–2.474 | 0.244 |
| BMI (kg/m²) (per 1 increase) | 0.950 | 0.903–0.999 | 0.046 | 1.077 | 1.003–1.157 | 0.004 |
| Cirrhosis (yes vs. no) | 0.776 | 0.452–1.335 | 0.360 | 0.456 | 0.164–1.272 | 0.134 |
| Child Pugh class (B or C vs. A) | 1.182 | 0.678–2.060 | 0.555 | 1.262 | 0.415–3.837 | 0.682 |
| AFP ng/ml, >200 vs. ≦200 | 3.351 | 2.181–5.148 | <0.001 | 2.246 | 1.233–4.093 | 0.008 |
| Tumor size (cm)>5cm (yes vs. no) | 5.030 | 2.996–8.447 | <0.001 | 1.028 | 0.452–2.340 | 0.947 |
| HCC diagnosis (Pathology vs. clinical) | 0.312 | 0.196–0.496 | <0.001 | 0.466 | 0.239–0.909 | 0.025 |
| 7th AJCC TNM stage | | | | | | |
| 2 vs. 1 | 1.363 | 0.525–3.537 | 0.525 | 0.648 | 0.214–1.963 | 0.443 |
| 3 vs. 1 | 8.081 | 3.810–17.139 | <0.001 | 2.560 | 0.772–8.487 | 0.124 |
| 4 vs. 1 | 10.487 | 4.850–22.677 | <0.001 | 4.874 | 1.462–15.651 | 0.01 |

BMI, body mass index; AJCC TNM, American Joint Committee on Cancer tumor-node-metastasis; BCLC, Barcelona Clinic Liver Cancer; AFP, alpha-feto protein; HCC, Hepatocellular carcinoma.

cancer-related symptoms with malnutrition.). Low-grade hepatic encephalopathy is difficult to differentiate from dementia, which is common in elderly patients [22]. The AASLD guidelines recommend the following: decompensated cirrhosis should be defined by the presence of "overt" clinical complications of cirrhosis (i.e. ascites, variceal hemorrhage, and hepatic encephalopathy) [23]. Relatedly, those patients with minimal ascites are not considered to have decompensated cirrhosis according the AASLD guideline recommendations [23]. In the HCC registry database of the hospital investigated in this study, only the Child-Pugh classes of the patients were recorded. The Child-Pugh score and the presence or absence of ascites, hepatic encephalopathy, and variceal hemorrhage were not recorded. Relatedly, there were stage 0 to stage C patients with minimal ascites who had Child-Pugh class A liver disease. Apart from the definition of decompensated cirrhosis being different between the two major international guidelines [6, 23], a previous study defined the presence of any ascites as portal hypertension rather than decompensated cirrhosis [24]. In summary, the definitions of compensated liver disease, clinically significant (relevant) portal hypertension, and decompensated cirrhosis need to be refined.

In the present study, 603 (33.5%) of the patients underwent LR, and only 65 (3.6%) of the patients underwent liver transplant. The investigated institution is the largest liver transplant center in Taiwan. However, while shortages of deceased liver donors have been noted worldwide, there is an extreme shortage of deceased liver donors in Taiwan relative to Western countries due to the local customs and traditions of Taiwan. Relatedly, the majority of patients who receive liver transplants in Taiwan receive living donor liver transplants [25]. A significant proportion of patients in the present study underwent LR because HBV is the leading etiology of HCC in Taiwan and a higher proportion of the patients had HBV–associated non-cirrhotic HCC. Our previous study found that around 60% of patients with HCC who underwent LR were non-cirrhotic [26]. Major liver resections can be performed with low rates of major complications in non-cirrhotic patients [6]. In contrast, cirrhosis underlies HCC in most patients and NAFLD is one of the leading etiologies of HCC in Western countries [7]. LR in patients with NAFLD and metabolic syndrome is burdened by a significant rate of severe complications [27, 28], and obesity-associated co-morbidities such as cardiovascular disease play a significant and negative prognostic role in NAFLD patients who undergo LR [27, 28]. In Taiwan, those with HBV, HCV, or cirrhosis of any etiology are reimbursed for HCC surveillance. 1085 (60.2%) of the patients in the present study were stage 0 or A patients. This high proportion of patients who were early-stage patients in the present study also raised the chances for LR being applied.

In the present study, upward treatment stage migration improved survival in selected stage B, C, and D patients, while downward treatment stage migration increased the risk of mortality in selected stage A but not stage 0 or stage C patients.

The tumor characteristics were more favorable and liver function reserve was better in the upward treatment stage migration group patients with BCLC stage B, C, and D HCC. Multivariate analysis showed that upward treatment stage migration was an independent and beneficial factor for OS in the BCLC stage B, C, and D patients. Relatedly, the better OS observed in the upward treatment stage migration group patients with BCLC stage B, C, and D HCC may have been due to selection bias and treatment effects.

For BCLC stage A patients, the tumor characteristics were more favorable and liver function reserve was better in the patients treated according to the BCLC guideline recommendations than in the patients in the downward treatment stage migration group. Furthermore, downward treatment migration was an independent and harmful factor associated with OS in the BCLC stage A patients. Relatedly, the worse OS observed in the downward treatment stage

migration group patients with BCLC stage A HCC may have been due to selection bias and treatment effects.

The characteristics of downward treatment stage migration and treatment according to the BCLC guideline recommendations were not significantly different in BCLC stage 0 patients. Furthermore, multivariate analysis showed that downward treatment stage migration was not an independent factor associated with OS in the BCLC stage 0 patients. For BCLC stage C patients, the liver function reserve was worse in the downward treatment stage migration group compared with the group treated according to the BCLC guideline recommendations. However, multivariate analysis showed that downward treatment stage migration was not an independent factor associated with OS in the BCLC stage C patients. Relatedly, we can conclude that downward treatment stage migration may not be associated with increased risk of mortality in selected stage 0 and stage C patients.

Notably, downward treatment stage migration [i.e. transcatheter arterial chemoemboliza-tion/embolization (TACE/TAE)] in BCLC stage 0 did not result in a worse survival rate. This result could be explained by TACE/TAE potentially having curative effects in selected cases (i.e. those in which an infarction area larger than the tumor is noted after TACE/TAE) and good effects in those with hypervascular and well-encapsulated tumors. Furthermore, the prognosis of patients with BCLC stage 0 may have been dictated by the severity of the liver disease rather than by the tumor status.

For the patients with BCLC stage C HCC, there was no significant difference in OS between those who received sorafenib (n = 93) and those who received downward treatment stage migration (i.e. best supportive care (BSC), n = 53; other treatments, n = 60) (p = 0.81). Other treatments used included systemic or hepatic artery infusion chemotherapy and external beam radiation therapy. Sorafenib treatment has been reimbursed by National Health Insurance system of Taiwan since August 2012. The criteria for reimbursement include major branch portal vein tumor thrombus (PVTT) (Vp3 or Vp4) [29] or extrahepatic spread (EHS) and Child-Pugh class A liver disease. Relatedly, patients with minor branch PVTT or mild deterioration of PS (i.e. PS 1 or 2) might receive other treatments. Those stage C patients who underwent BSC were categorized as receiving downward treatment stage migration, which means that the stage C patients who received downward treatment stage migration were composed of both better and worse patients compared with those who underwent sorafenib treatment. This could explain why there was no significant difference in OS between the two groups.

The EASL guidelines proposed the concept of treatment stage migration [6]. For example, patients at BCLC stage B with contraindications for TACE [30, 31] should be offered sorafenib, as reported in the SHARP trial [32]. However, sorafenib treatment was not reimbursed for *de novo* BCLC stage B patients in Taiwan. Relatedly, there were only 2 patients in BCLC stage B who underwent downward treatment stage migration with sorafenib in the present study.

Non-adherence to the modified BCLC staging system was common for the stage B and C patients in the present study. In those with stage B HCC, TAE/TACE was still the most common treatment modality (49.6%), followed by LR (33.9%). The BCLC stage B is heterogeneous, and whether LR is better than TACE for BCLC stage B is still an unsolved question [33]. In those with stage C, sorafenib was the most common treatment modality (26.0%), follow by LR (17.0%), other treatments (16.8%), and BSC (14.8%). The EASL guidelines recommend that LR only be considered for minor branch PVTT of HCC (Vp1 or Vp2) [6]. At the investigated hospital, LR can be considered for PVTT limited to the first-order branch (Vp3) [29], and the outcomes are acceptable, as reported in our previous study [34]. A significant proportion of the patients with stage C HCC in this study underwent other treatments because sorafenib was reimbursed for selected stage C patients.

The present study included a cohort of HCC patients enrolled over 7 recent years. Relatedly, the diagnosis and treatment of HCC for these patients were performed according to the state of the art and under MDT discussions. The study group featured a large proportion of patients with early-stage HCC. Finally, the patients were selected prospectively and prognostic staging was done for each patient before treatment commenced. The present study is the first study to prospectively validate the prognostic capability of the modified BCLC staging system. Previous studies excluded patients who underwent liver transplant and reanalyzed the prognostic capability of the original BCLC staging system with the aim of eliminating the beneficial effects conferred by removal of a cirrhotic liver [24, 35]. In the present study, we did not reanalyze the data in this way because few of the patients in this cohort underwent liver transplant.

The strength of the present study was the complete collection of data for a large number of patients. The patients were enrolled over a recent and short period of time, meaning that the diagnosis and treatment of HCC for them were more consistent. Second, we checked the vital statuses of the patients by using an authoritative source. We could thus be sure of the vital statuses of the patients enrolled in the study. The limitations of the present study were as follows: First, the modified BCLC guidelines proposed that Child-Pugh A without any ascites is essential for all stages except stage D. However, the presence or absence of ascites is not recorded in the HCC registry database of the hospital in question. The modified BCLC guidelines recommend sorafenib for stage C patients and those with stage B HCC not eligible for TACE. However, reimbursement for sorafenib in the study period was restricted only to those with major branch PVTT or EHS. Second, this was a single-center study. The investigated institution is a high-volume liver surgery center, and the results may thus not be generalizable to low-volume liver surgery institutions. Our results may also not apply to patients with HCC in Western countries because of differences in ethnicity, the etiologies of liver disease, the proportions of patients with cirrhosis, and, most importantly, health care systems. Third, the number of patients in this study who received BSC may have been underestimated. The HCC registry database of the investigated hospital records the data of patients who were diagnosed and managed at the institution. Therefore, those who were diagnosed at our institution and received treatment (e.g. BSC) at another hospital were not enrolled. Fourth, the BCLC guidelines recommend that the deterioration of PS is related to cancer-related symptoms. However, 19 (16.4%) of the patients with stage D HCC in this study underwent liver transplant. The indications for liver transplant used at the investigated hospital are within the University of California San Francisco (UCSF) criteria [36]. Relatedly, these patients with a low tumor burden should not have cancer-related symptoms. However, as noted by Giannini et al., the deterioration of PS is very subjective. Several confounding factors, such as being elderly, having advanced cirrhosis, and having severe comorbidities may play a major role [37]. Finally, for patients who were still alive after December 2018 and more than 5 years since the date of HCC diagnosis, the last follow-up date would be 5 years since the date of HCC diagnosis. Relatedly, the follow-up periods of patients who were still alive after December 2018 and more than 5 years since the date of HCC diagnosis would be underestimated. The median survival in the current study may thus be underestimated.

In conclusion, this study reports the adherence to the modified BCLC guidelines at a high-volume liver surgery center in Taiwan. The unique health care system in Taiwan is totally different from those in Western countries. Relatedly, there was no referral bias in the present study. The patients enrolled in the present study could thus be representative of the general HCC population in Taiwan. Few of the patients underwent liver transplant because of local customs and traditions leading to an extreme shortage of deceased donors. Furthermore, a significant proportion of patients underwent LR in this high-volume liver surgery center because of the higher prevalence of HBV-associated non-cirrhotic HCC, while the majority of the

present cohort were early-stage HCC patients because of an HCC surveillance program. Non-adherence to the modified BCLC staging system was common in treating the stage B and C patients in this study. Furthermore, using prospectively collected data, we have validated the prognostic capability of the modified BCLC staging system, which had not previously been prospectively validated.

## Supporting information

**S1 Raw data. Raw data of this cohort.**
(XLSX)

## Acknowledgments

The authors thank Cancer Center, Kaohsiung Chang Gung Memorial Hospital for the provision of HCC registry data. The authors thank Chih-Yun Lin and Nien-Tzu Hsu and the Biostatistics Center, Kaohsiung Chang Gung Memorial Hospital for statistics work.

## Author Contributions

**Conceptualization:** Yi-Hao Yen.

**Data curation:** Yi-Hao Yen.

**Formal analysis:** Yi-Hao Yen.

**Funding acquisition:** Yi-Hao Yen.

**Investigation:** Yi-Hao Yen.

**Methodology:** Yi-Hao Yen.

**Project administration:** Yi-Hao Yen.

**Resources:** Yi-Hao Yen.

**Software:** Yi-Hao Yen.

**Supervision:** Yi-Hao Yen, Yu-Fan Cheng, Jing-Houng Wang, Chih-Che Lin, Chien-Hung Chen, Chih-Chi Wang.

**Validation:** Yi-Hao Yen.

**Visualization:** Yi-Hao Yen.

**Writing – original draft:** Yi-Hao Yen.

**Writing – review & editing:** Yi-Hao Yen.

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
