## [Decision Letter · Decision Letter 0]

11 Feb 2021

PONE-D-20-22087

Adherence to the modified Barcelona Clinic Liver Cancer guidelines: Results from a high-volume liver surgery center in East Asia

PLOS ONE

Dear Dr. Yen,

Thank you for submitting your manuscript to PLOS ONE. After careful consideration, we feel that it has merit but does not fully meet PLOS ONE’s publication criteria as it currently stands. Therefore, we invite you to submit a revised version of the manuscript that addresses the points raised during the review process.

We look forward to receiving your revised manuscript.

Kind regards,

Wen-Chi Chou

Academic Editor

PLOS ONE

Journal Requirements:

2)  Thank you for stating the following financial disclosure:

 [The funders had no role in study design, data collection and analysis, decision to

publish, or preparation of the manuscript.].

3) Please include captions for your Supporting Information files at the end of your manuscript, and update any in-text citations to match accordingly. Please see our Supporting Information guidelines for more information: http://journals.plos.org/plosone/s/supporting-information.

Reviewers' comments:

Reviewer's Responses to Questions

**Comments to the Author**

1. Is the manuscript technically sound, and do the data support the conclusions?

Reviewer #1: Partly

Reviewer #2: Partly

2. Has the statistical analysis been performed appropriately and rigorously? 

Reviewer #1: Yes

Reviewer #2: Yes

3. Have the authors made all data underlying the findings in their manuscript fully available?

Reviewer #1: Yes

Reviewer #2: No

4. Is the manuscript presented in an intelligible fashion and written in standard English?

Reviewer #1: Yes

Reviewer #2: Yes

5. Review Comments to the Author

Reviewer #1: This is a retrospective study of HCC patients at a very busy liver surgery center. Their state purpose is to describe the patients with HCC and determine whether adherence to the modified BCLC guidelines impact patient outcomes.

The group has quite an impressive clinical volume and experience with HCC.

For the Introduction, I think the paper would be more clear with additional background information. Both the abstract and the intro jump into the nuances of the BCLC without giving much context. Some of the background in the discussion might be moved up here to make it easier to understand for people not immersed in this all the time or learning about it for the first time. The purpose, as stated of this research is also vague. Perhaps more context can make it more clear.

For the Methods section, the large single center experience enabled acquisition of a very complete dataset. However, the major weakness is that this is a non-randomized study, from a single center, and its hard to determine why patients were treated differently. Therefore, it make it difficult to associate outcome.

Since the purpose of this research is somewhat vague it is hard to focus on the important point through the results section and the discussion. As mentioned earlier, moving some of the intial discussion up to the introduction may set up the rest of the paper better.

Reviewer #2: The author provided a large number of HCCs in high volume center to validate the modified BCLC stage/treatment, but it needs more information/analysis in survival to prove author’s conclusion.

1. Please provide clinical characteristics of adherence/non-adherence group in each BCLC stage. (include treatment options, follow up time..etc)

2. Explain the survival curve in fig 2. If there are some mistake in survival analysis, please modify it. For example(a) all patient died in 72month? (b)BCLC C had better survival than D?

3. suggest to do multivariable analysis in the group which patient number is enough, ex fig. 3B

4. In Fig 3C, the patient number in downward treatment is small, it is difficult to support the conclusion

5. In 3D and 3E, the survival curve is too good in upward treatment group, please explain.

6. PLOS authors have the option to publish the peer review history of their article (what does this mean?). If published, this will include your full peer review and any attached files.

Reviewer #1: No

Reviewer #2: **Yes: **Chien hui Wu

---

## [Author Response · Author response to Decision Letter 0]

9 Mar 2021

Reviewer #1: This is a retrospective study of HCC patients at a very busy liver surgery center. Their state purpose is to describe the patients with HCC and determine whether adherence to the modified BCLC guidelines impact patient outcomes.

The group has quite an impressive clinical volume and experience with HCC.

For the Introduction, I think the paper would be more clear with additional background information. Both the abstract and the intro jump into the nuances of the BCLC without giving much context. Some of the background in the discussion might be moved up here to make it easier to understand for people not immersed in this all the time or learning about it for the first time. The purpose, as stated of this research is also vague. Perhaps more context can make it more clear.

Response: We have revised the manuscript according to your comments. 

Abstract 

Background and aims: The Barcelona Clinic Liver Cancer (BCLC) staging system is the most widely applied staging system for hepatocellular carcinoma (HCC) and is recommended for treatment allocation and prognostic prediction. The BCLC guidelines were modified in 2018 to indicate that Child-Pugh A without any ascites is essential for all stages except stage D. This study sought to provide a description of patients with HCC treated at a high-volume liver surgery center in Taiwan where referral is not needed and all treatment modalities are available and reimbursed by the National Health Insurance program. As such, certain variables that could modulate treatment decisions in clinical practice, including financial constraints, the availability of treatment procedures, and the expertise of the hospital, could be excluded. The study further sought to evaluate the adherence to the modified BCLC guidelines.

Methods: This was a retrospective study with prospectively collected data. 1801 consecutive patients with de novo HCC were enrolled through our institution from 2011-2017.

Results: There were 302 patients with stage 0, 783 with stage A, 242 with stage B, 358 with stage C, and 116 with stage D HCC. Treatment adhering to the modified BCLC guidelines recommendations was provided to 259 (85.8%) stage 0 patients, 606 (77.4%) stage A patients, 120 (49.6%) stage B patients, 93 (26.0%) stage C patients, and 83 (71.6%) stage D patients. 

Conclusions: We reported treatment adhering to the modified BCLC guidelines at a high-volume liver surgery center in Taiwan. We found that non-adherence to the modified BCLC staging system was common in treating stage B and C patients. 

Please see page 3-4.

Introduction

Hepatocellular carcinoma (HCC) is one of the leading causes of cancer death worldwide [1,2]. In patients with HCC tumors, unlike other solid tumors, the co-existence of two life-threatening diseases (i.e. cancer and liver cirrhosis) complicates the prognostic evaluation [3, 4]. The Barcelona Clinic Liver Cancer (BCLC) staging system includes tumor characteristics, liver function reserve (i.e. Child-Pugh class), and performance status; it is the most widely applied staging system for HCC and is recommended for treatment allocation and prognostic prediction [5]. Please see page 5, first paragraph. 

Financial constraints play a pivotal role in adherence to the modified BCLC guideline recommendations. The current healthcare system in Taiwan, known as the National Health Insurance system, is totally different from those in Western countries. The system has covered more than 99% of Taiwan’s citizens since 2004. Under the system, citizens are free to choose physicians and hospitals without referral, and all the treatment modalities for HCC are reimbursed. Furthermore, patients with cancers can apply for a catastrophic illness card. Therefore, patients with HCC do not have to pay anything when they receive medical care related to HCC. Nearly 90% of citizens are satisfied with the current health care system in Taiwan (https://www.mohw.gov.tw/cp-4251-50316-1.html). Please see page 6, last paragraph and page 7, first paragraph. 

For the Methods section, the large single center experience enabled acquisition of a very complete dataset. However, the major weakness is that this is a non-randomized study, from a single center, and its hard to determine why patients were treated differently. Therefore, it make it difficult to associate outcome.

Response: Thank you so much for your comments. 

We have provided the characteristics of upward or downward treatment stage migration and treatment according to the BCLC guideline recommendations in each stage. Furthermore, we have performed multivariate analysis to evaluate the influence on overall survival of non-adherence to the BCLC guideline recommendations in each BCLC stage. 

In the present study, upward treatment stage migration improved survival in selected stage B, C, and D patients, while downward treatment stage migration increased the risk of mortality in selected stage A but not stage 0 or stage C patients. The tumor characteristics were more favorable and liver function reserve was better in the upward treatment stage migration group patients with BCLC stage B, C, and D HCC. Multivariate analysis showed that upward treatment stage migration was an independent and beneficial factor for OS in the BCLC stage B, C, and D patients. Relatedly, the better OS observed in the upward treatment stage migration group patients with BCLC stage B, C, and D HCC may have been due to selection bias and treatment effects. 

For BCLC stage A patients, the tumor characteristics were more favorable and liver function reserve was better in the patients treated according to the BCLC guideline recommendations than in the patients in the downward treatment stage migration group. Furthermore, downward treatment migration was an independent and harmful factor associated with OS in the BCLC stage A patients. Relatedly, the worse OS observed in the downward treatment stage migration group patients with BCLC stage A HCC may have been due to selection bias and treatment effects.

The characteristics of downward treatment stage migration and treatment according to the BCLC guideline recommendations were not significantly different in BCLC stage 0 patients. Furthermore, multivariate analysis showed that downward treatment stage migration was not an independent factor associated with OS in the BCLC stage 0 patients. For BCLC stage C patients, the liver function reserve was worse in the downward treatment stage migration group compared with the group treated according to the BCLC guideline recommendations. However, multivariate analysis showed that downward treatment stage migration was not an independent factor associated with OS in the BCLC stage C patients. Relatedly, we can conclude that downward treatment stage migration may not be associated with increased risk of mortality in selected stage 0 and stage C patients. Please page 62, 2nd paragraph to page 63, first and 2nd paragraph. 

Since the purpose of this research is somewhat vague it is hard to focus on the important point through the results section and the discussion. As mentioned earlier, moving some of the initial discussion up to the introduction may set up the rest of the paper better.

Response: Thank you so much for your comments. We have now moved some of the initial discussion up to the introduction.

Reviewer #2: The author provided a large number of HCCs in high volume center to validate the modified BCLC stage/treatment, but it needs more information/analysis in survival to prove author’s conclusion.

1. Please provide clinical characteristics of adherence/non-adherence group in each BCLC stage. (include treatment options, follow up time..etc)

Response: Thank you so much for your comments. We have now provided the clinical characteristics of adherence/non-adherence groups in each BCLC stage. 

Characteristics of adherence and non-adherence to BCLC guideline recommendations in each BCLC stage

BCLC stage 0

Among BCLC stage 0 patients, there was a higher proportion of patients diagnosed by pathology, and the follow-up period was shorter in the patients treated according to the BCLC guideline recommendations than in the downward treatment stage migration group. There were no significant difference between the two groups in terms of other characteristics (Table 3). 

BCLC stage A

Among BCLC stage A patients, the mean age was lower, the proportion of patients with cirrhosis was lower, the bilirubin and INR levels were lower, and the proportion of HCCs diagnosed by pathology and with TNM stage 1 was higher in the patients treated according to the BCLC guideline recommendations than in the downward treatment stage migration group. There were no significant differences between the two groups in terms of other characteristics (Table 4).

BCLC stage B

Among BCLC stage B patients, the proportion of patients with cirrhosis was lowest, the bilirubin and INR levels were lowest, and the proportion of HCCs diagnosed by pathology and with TNM stage 1 was highest in the upward treatment migration group compared with the other groups. Meanwhile, the proportion of patients with Child-Pugh class A was highest and the follow-up period was longer in the patients treated according to the BCLC guideline recommendations than in the other patient groups. There were no significant differences in terms of other characteristics among the groups (Table 5).

BCLC stage C

Among BCLC stage C patients, the mean age was highest, the proportions of male patients and those with tumor size >5.0 cm were lowest, the proportion of patients with AFP<20 ng/ml was highest, the proportion of HCCs diagnosed by pathology and with TNM stage 1 was highest, the bilirubin and INR levels were lowest, and the follow-up period was longer in the upward treatment stage migration group than in the other patient groups. The proportion of patients with Child-Pugh class A was highest in the patients treated according to the BCLC guideline recommendations compared with the other patient groups. There were no significant difference in terms of other characteristics among the groups (Table 6). 

BCLC stage D

Among BCLC stage D patients, the mean age was younger, the BMI was higher, the proportion of patients with Child-Pugh class A was higher, the proportion of patients with AFP< 20ng/ml was higher, the proportion of HCCs diagnosed by pathology and with TNM stage 1 was higher, the creatinine and bilirubin levels were lower, the proportion of patients with tumor size > 5.0 cm was lower and the follow-up period was longer in the upward treatment stage migration group than in the patients treated according to the BCLC guideline recommendations. There were no significant difference between the groups in terms of other characteristics (Table 7). Please see page 23-45. 

2. Explain the survival curve in fig 2. If there are some mistake in survival analysis, please modify it. For example(a) all patient died in 72month? (b)BCLC C had better survival than D?

Response: Thank you so much for your comments.

(a). The following text is now included in the method section: 

Data were extracted from the Kaohsiung Chang Gung Memorial Hospital HCC registry database, including the data for 1801 de novo HCC patients consecutively evaluated and managed at the hospital from January 2011 to December 2017. A flow chart of the patients’ enrollment is shown in Fig. 1. The data contained in the HCC registry database of the hospital were prospectively collected. The vital status of every single HCC patient was updated annually by linking to the website of the Ministry of Health and Welfare of Taiwan (https://hosplab.hpa.gov.tw/CSTIIS/index.aspx). The last update of vital statuses performed by linking to the website of the Ministry of Health and Welfare of Taiwan (https://hosplab.hpa.gov.tw/CSTIIS/index.aspx) in the current study was conducted in December 2018. The personnel who registered the HCC registry data checked the last follow-up date for each patient at 1, 3, and 5 years after the date of HCC diagnosis. The last follow-up date for each patient would be the last visit to our hospital, except for those patients who did not have follow-up visits at our hospital, who were contacted by phone. The last follow-up date was checked using this method until 5 years after the date of HCC diagnosis. If patients were still alive after December 2018 and more than 5 years since the date of HCC diagnosis, the last follow-up date would be 5 years since the date of HCC diagnosis. The method used to check the vital statuses and last follow-up dates of patients in the current study was different from performing manual medical record reviews. To avoid causing any confusion, we showed the 5-year overall survival (OS) rates in all figures in this study. Please see page 8, last paragraph and page 9, first paragraph. 

In the discussion, we have acknowledged this point in discussing the limitations of the current study: Finally, for patients who were still alive after December 2018 and more than 5 years since the date of HCC diagnosis, the last follow-up date would be 5 years since the date of HCC diagnosis. Relatedly, the follow-up periods of patients who were still alive after December 2018 and more than 5 years since the date of HCC diagnosis would be underestimated. The median survival in the current study may thus be underestimated. Please see page 68, 2nd paragraph. 

(b). BCLC C had better survival than D, log rank test stage C vs. D, p = 0.015.

3.suggest to do multivariable analysis in the group which patient number is enough, ex fig. 3B

Response: Thank you so much for your comments. We have now provided the results of multivariable analysis for each BCLC stage. 

To evaluate the impact of non-adherence to the BCLC guideline recommendations on OS, the characteristics of patients with upward treatment stage migration and downward treatment stage migration according to the BCLC guideline recommendations were compared by chi-square test, Fisher’s exact test, Mann-Whitney U test, Kurskal-Wallis test, one-way ANOVA, and independent-samples t test, as appropriate. The variables enrolled for univariate analysis were: age, sex, diagnosis method of HCC (pathology vs. clinical), tumor size, AJCC TNM stage, body mass index (BMI), etiology of liver disease, AFP level, cirrhosis status, Child-Pugh class, creatinine level, bilirubin level, and international normalized ratio (INR). 

The Cox model was adjusted for those variables with a p-value <0.05 or clinical relevance (e.g. AFP level). To avoid collinearity, redundant variables were not enrolled in the multivariate analysis. The reason why we enrolled the diagnosis method of HCC (pathology vs. clinical) was because if a patient’s HCC was diagnosed clinically, it meant that the patient underwent non-surgical treatment. Furthermore, LR may improve survival across BCLC stages [11]. Please see page 12, 2nd paragraph. 

Predictors of OS by univariate and multivariate analysis

Among BCLC stage 0 patients, INR (HR=5.189, 95%CI=1.933-13.933, p=0.001) was the only independent factor associated with OS in the multivariate analysis, while treatment adherence vs. non-adherence was not.

Among BCLC stage A patients, downward treatment stage migration (HR=1.881, 95%CI=1.342-2.638, p<0.001), age > 65 years (HR=1.369, 95%CI=1.041-1.801, p=0.025), cirrhosis status (HR=1.539, 95%CI=1.081-2.193, p=0.017), and AFP level > vs. ≦200 ng/ml (HR=1.804, 95%CI=1.321-2.464, p<0.001) were independent predictors for OS.

Among BCLC stage B patients, upward treatment stage migration (HR=0.413, 95%CI=0.246-0.693, p=0.001), Child-Pugh class B vs. A (HR=4.656, 95%CI=1.343-16.145, p=0.015), AFP level > vs. ≦200 ng/ml (HR=2.799, 95%CI=1.891-4.142, p<0.001), and TNM stage 3 vs. 1 or 2 (HR=2.865, 95%CI=1.905-4.308, p<0.001) were independent predictors for OS.

Among BCLC stage C patients, upward treatment stage migration (HR=0.464, 95%CI=0.324-0.663, p<0.001), Child-Pugh class B or C vs. A (HR=3.096, 95%CI=2.060-4.654, p<0.001), HCC diagnosed by pathology (HR=0.659, 95%CI=0.505-0.860, p=0.002), and TNM stage 4 vs. 1 (HR=2.731, 95%CI=1.460-5.110, p=0.002) were independent predictors for OS.

Among BCLC stage D patients, upward treatment stage migration (HR=0.144, 95%CI=0.043-0.483, p=0.002), BMI per 1 unit increase (HR=1.077, 95%CI=1.003-1.157, p=0.004), AFP level > vs. ≦200 ng/ml (HR=2.246, 95%CI=1.233-4.093, p=0.008), HCC diagnosed by pathology vs. clinical (HR=0.466, 95%CI=0.239-0.909, p=0.025), and TNM stage 4 vs. 1 (HR=4.874, 95%CI=1.462-15.651, p=0.01) were independent predictors for OS. Please page 46-59. 

4. In Fig 3C, the patient number in downward treatment is small, it is difficult to support the conclusion

Response: Thank you so much for your comments. We did not mention that downward treatment stage migration was associated with worse OS in BCLC stage B patients. 

5. In 3D and 3E, the survival curve is too good in upward treatment group, please explain.

Response:　 Thank you so much for your comments.

Characteristics of adherence and non-adherence to BCLC guideline recommendations in each BCLC stage

BCLC stage C

Among BCLC stage C patients, the mean age was highest, the proportions of male patients and those with tumor size >5.0 cm were lowest, the proportion of patients with AFP<20 ng/ml was highest, the proportion of HCCs diagnosed by pathology and with TNM stage 1 was highest, the bilirubin and INR levels were lowest, and the follow-up period was longer in the upward treatment stage migration group than in the other patient groups. The proportion of patients with Child-Pugh class A was highest in the patients treated according to the BCLC guideline recommendations compared with the other patient groups. There were no significant difference in terms of other characteristics among the groups (Table 6). 

BCLC stage D

Among BCLC stage D patients, the mean age was younger, the BMI was higher, the proportion of patients with Child-Pugh class A was higher, the proportion of patients with AFP< 20ng/ml was higher, the proportion of HCCs diagnosed by pathology and with TNM stage 1 was higher, the creatinine and bilirubin levels were lower, the proportion of patients with tumor size > 5.0 cm was lower and the follow-up period was longer in the upward treatment stage migration group than in the patients treated according to the BCLC guideline recommendations. There were no significant difference between the groups in terms of other characteristics (Table 7). Please see page 36-45.

Predictors of OS by univariate and multivariate analysis

Among BCLC stage C patients, upward treatment stage migration (HR=0.464, 95%CI=0.324-0.663, p<0.001), Child-Pugh class B or C vs. A (HR=3.096, 95%CI=2.060-4.654, p<0.001), HCC diagnosed by pathology (HR=0.659, 95%CI=0.505-0.860, p=0.002), and TNM stage 4 vs. 1 (HR=2.731, 95%CI=1.460-5.110, p=0.002) were independent predictors for OS (Table 11).

Among BCLC stage D patients, upward treatment stage migration (HR=0.144, 95%CI=0.043-0.483, p=0.002), BMI per 1 unit increase (HR=1.077, 95%CI=1.003-1.157, p=0.004), AFP level > vs. ≦200 ng/ml (HR=2.246, 95%CI=1.233-4.093, p=0.008), HCC diagnosed by pathology vs. clinical (HR=0.466, 95%CI=0.239-0.909, p=0.025), and TNM stage 4 vs. 1 (HR=4.874, 95%CI=1.462-15.651, p=0.01) were independent predictors for OS (Table 12). Please page 54-59. 

The tumor characteristics were more favorable and liver function reserve was better in the upward treatment stage migration group patients with BCLC stage B, C, and D HCC. Multivariate analysis showed that upward treatment stage migration was an independent and beneficial factor for OS in the BCLC stage B, C, and D patients. Relatedly, the better OS observed in the upward treatment stage migration group patients with BCLC stage B, C, and D HCC may have been due to selection bias and treatment effects. Please see page 62, last paragraph.

---

## [Decision Letter · Decision Letter 1]

15 Mar 2021

Adherence to the modified Barcelona Clinic Liver Cancer guidelines: Results from a high-volume liver surgery center in East Asia

PONE-D-20-22087R1

Dear Dr. Yen,

We’re pleased to inform you that your manuscript has been judged scientifically suitable for publication and will be formally accepted for publication once it meets all outstanding technical requirements.

Kind regards,

Wen-Chi Chou

Academic Editor

PLOS ONE

Additional Editor Comments (optional):

Reviewers' comments:

Reviewer's Responses to Questions

**Comments to the Author**

1. If the authors have adequately addressed your comments raised in a previous round of review and you feel that this manuscript is now acceptable for publication, you may indicate that here to bypass the “Comments to the Author” section, enter your conflict of interest statement in the “Confidential to Editor” section, and submit your "Accept" recommendation.

Reviewer #2: All comments have been addressed

2. Is the manuscript technically sound, and do the data support the conclusions?

Reviewer #2: Yes

3. Has the statistical analysis been performed appropriately and rigorously? 

Reviewer #2: Yes

4. Have the authors made all data underlying the findings in their manuscript fully available?

Reviewer #2: Yes

5. Is the manuscript presented in an intelligible fashion and written in standard English?

Reviewer #2: Yes

6. Review Comments to the Author

Reviewer #2: (No Response)

7. PLOS authors have the option to publish the peer review history of their article (what does this mean?). If published, this will include your full peer review and any attached files.

Reviewer #2: **Yes: **Chien hui Wu

---

## [Editor Report · Acceptance letter]

17 Mar 2021

PONE-D-20-22087R1 

Adherence to the modified Barcelona Clinic Liver Cancer guidelines: Results from a high-volume liver surgery center in East Asia 

Dear Dr. Yen:

I'm pleased to inform you that your manuscript has been deemed suitable for publication in PLOS ONE. Congratulations! Your manuscript is now with our production department. 

Kind regards, 

on behalf of

Dr. Wen-Chi Chou 

Academic Editor

PLOS ONE